# Effects of psychogenic stress on oxidative stress and antioxidant capacity at different growth stages of rats: Experimental study

Sangun Lee[1,2]*, Atsunori Itagaki[1], Atsuko Satoh[3], Issei Sugimoto[2], Takumi Saito[2], Yoshihiko Shibukawa[2], Haruka Tatehana[4]

1 Department of Physical Therapy, Aomori University of Health and Welfare, Aomori-shi, Aomori-ken, Japan, 2 Aomori University of Health and Welfare Graduate School of Health Sciences, Aomori-shi, Aomori-ken, Japan, 3 Faculty of Nursing, Hirosaki Gakuen University, Hirosaki-shi, Aomori-ken, Japan, 4 Department of Nutrition, Aomori University of Health and Welfare, Aomori-shi, Aomori-ken, Japan

* leesu@auhw.ac.jp

## Abstract

This study examined the psychogenic stress (PS) effects on changes in oxidative stress and the antioxidant capacity of an organism at different growth stages. The experimental animals were male Wistar rats of five different ages from growth periods (GPs) to old age. The growth stages were randomly classified into control (C) and experimental (PS) groups. The PS was performed using restraint and water immersion once daily for 3 h for 4 weeks. Reactive oxygen metabolites (d-ROMs) and the biological antioxidant potential (BAP) were measured before and after the experiment. In addition, the liver and adrenal glands were removed, and the wet weight was measured. The d-ROM and BAP of all growth stages given PS increased significantly. The d-ROM in the C group without PS increased significantly in GPs while decreased significantly in old-aged rats. In addition, the BAP of the C group in GP and early adulthood were all significantly elevated. There were significant differences in organ weights between the C and PS groups at all growth stages. Oxidative stress and antioxidant capacity differed depending on the organism's developmental status and growth stage, and PS also showed different effects. In particular, the variability in oxidative stress was remarkable, suggesting that the effect of PS was more significant in the organism's immature organs.

## Introduction

Unfamiliarity with the living environment and sudden changes in living conditions can promote psychogenic stress (PS), leading to various diseases [1,2]. In recent years, the coronavirus disease (COVID-19) pandemic has caused various restrictions in daily life, and there have been concerns about its impact on the body [3–5].

Reactive oxygen species (ROS) play an important role in primary immune responses *in vivo* [6,7]. However, excessive accumulation of ROS *in vivo* results in highly toxic oxidative stress that damages normal cells [8,9]. Therefore, an appropriate level of oxidative stress *in vivo* is

and antioxidant capacity at different growth stages of rats: Experimental study" by Sangun Lee.

**Funding:** This research was supported by Health promotion "Research Center for Strategy of Health Promotion" of Aomori University of health and welfare (Grant No. 2020-5, 2021-1).

**Competing interests:** The authors declare no conflicts of interest associated with this manuscript.

required to maintain healthy organisms. ROS accounts for 0.1–0.2% of oxygen consumption in mitochondria involved in energy metabolism [10] and increases markedly with aging and anaerobic physical activity [11]. Oxidative stress *in vivo* is regulated by antioxidant enzymes such as super oxide dismutase (SOD) and glutathione peroxidase (GPx) to maintain stable oxidative stress levels *in vivo* at all times [8,9].

Stress also has negative effects on organs, such as adrenal enlargement and atrophy of the thymus gland, and causes the disruption of homeostasis and dysfunction of the endocrine system and immune cells [12,13]. Herrera-Covarrubias et al. [14] and Araujo et al. [15] also reported stress-induced effects on hormones, suggesting the involvement of stress in hormone function. Furthermore, Kuhn CM et al. [16] reported the effects of stress on growth and immune function, suggesting a reduction in metabolic levels of energy production and cellular response to growth hormones owing to stress. Differences in organ function and hormonal changes during the organism's developmental process and aging may result in different responses to oxidative stress and antioxidant capacity, and PS may further contribute to these responses. However, the effects of PS on oxidative stress and antioxidant capacity dynamics have not been clarified, and the characteristics of the biological responses during the growth stages are unknown.

In this study, PS was administered to organisms at different growth stages to clarify the variation in oxidative stress and antioxidant capacity and to obtain basic knowledge for preventive medicine in response to oxidative stress.

## Methods

All procedures were approved by the institutional animal care and use committee of the Aomori University of Health and Welfare (AUHW) (Animal ethics number: 20001).

### Experimental animals

The experimental animals were male Wistar rats (CLEA Japan, Inc. Japan) at five different growth stages: Growth periods (GPs, 3 weeks), early adulthood (EA, 8 weeks), adulthood (A, 6 months), middle-aged (MA, 12 months), and old-aged (OA, 24 months) (n = 12, for each stage). The animals were randomly divided into control (C) and experimental (PS) groups for each growth stage. The number of animals in each group was set at six to account for mortality during the experiment and stress caused by the rearing environment. All experimental animals were housed in the experimental animal room at AUHW from 3 weeks of age until the day of each experiment. Considerations were made for the effects of the rearing environment, such as room temperature and food, on the animals. The rearing environment was two animals in one cage at a room temperature of 22 ± 2˚C, a humidity of 55 ± 5%, and a 12-h light-dark cycle. The animals were allowed to move freely in their cages and engage in physical activity. They were also allowed to consume food CE-2 (CLEA Japan, Inc. Japan) and water ad libitum during the experiment.

### Protocol of psychogenic stress

The restraint water-immersion stress (RWIS) method used in previous studies by Lu et al. [17] and Ohta et al. [18,19] was used for PS. The RWIS method is a PS method that takes advantage of the water-hating nature of rats and physical restraint. The 3 h RWIS experiment by Ota et al. [18,19] showed an increase in serum lipid peroxide (LPO) and nitrite/nitrate (NOx) concentrations and a decrease in brain ascorbate and glutathione content. PS was performed once a day for 3 h at room temperature (22 ± 2˚C) and humidity (55 ± 5%), similar to those in the animal breeding environment. To examine the morphological development and changes in

organs owing to RWIS in this experiment, we considered a 4-week period from growth periods to early adulthood. All experiments were conducted simultaneously over 4 weeks. The RWIS used small animal fixation adjusters (Sanplatec. co. Japan) with adjustable lengths to account for individual differences in age and body weight (body size). Rats were placed in the "rat fixation adjuster" that adjusts to three sizes of experimental rats (small: Φ31 x 20~80 mm, medium: Φ60 x 60~170 mm, large: Φ76 x 70~220 mm). The intake port at the front of the fixation adjuster was moved to the tail to the maximum setting, and the rats were restrained. The restrained rats were then transferred to a tank with a 2-cm water level and submerged (water temperature of 23˚C). Since the age of this study's subjects ranged from youth to old age, fasting was not performed to eliminate the effects of food stress.

## Blood collection and anesthesia

The experimental animals were weighed using a digital scale both before and after the experiment. The experimental animals were immobilized using a rat fixation adjuster both before and after the experiment, and 1 ml of blood was collected from the tail vein, respectively. The blood samples were centrifuged at 3,000 rpm for 15 minutes. The centrifuged serum was then transferred into micro tubes and stored in a deep freezer at −80 ˚C until analysis.

The anesthesia was prepared with 25 ml of three types of mixed anesthetic agents (Medetomidine Hydrochloride 7.5μg; 0.375 mg/kg + Midazolam 0.4 mg; 2.0 mg/kg + Butorphanol Tartrate 0.5 mg;2.5 mg/kg) and administered intraperitoneally (0.5 ml / BW100 g) after blood collection at the end of the experiment.

## Organ evisceration

The chest was opened after confirming that the animal was under deep anesthesia. The animal was perfused with saline from the left ventricle, and the liver and right adrenal gland were immediately removed. These were then immersed in 10% neutral buffered formalin solution and stored until analysis. For analysis, they were washed in saline, trimmed, and wiped dry before wet weight was determined.

## Oxidant stress and antioxidant capacity

To assess oxidative stress and antioxidant potential, reactive oxygen metabolites (d-ROMs) and biological antioxidant potential (BAP) were measured in the plasma. The d-ROMs and BAP were measured using REDOXLIBRA (WISMERLL Co, Ltd., Japan) [11,20,21].

**Oxidative stress (reactive oxygen metabolites: d-ROMs).** The d-ROMs quantifies the amount of serum hydroperoxide (R-OOH), a metabolite of reactive oxygen species and free radicals. The measurement relies on a colorimetric method to react divalent iron (Fe2+) and trivalent iron (Fe3+). Then, the coloring solution chromogen (N,N diethyl paraphenylenediamine) is applied to the alkoxyl radical (R-O.) and the peroxy radical (R-OO.). The resulting final substance (A-NH2) is obtained by measuring. The main reactions in the d-ROM measurement are shown; "R-OOH + Fe2+ → R-O- + Fe3+ + OH-R-O- + A-NH2 → R-O- + [A-NH2-]+", "R-OOH + Fe3+ → R-OOH- + Fe2+ + H+R-OOH- + A-NH2 → R-OOH- + [A-NH2-]+".

The assay method was as follows: The d-ROMs assay kit cuvette (4.8PH acetate buffer) is placed in a thermostat space set at 37˚C and allowed to warm for 10 minutes. Then, 20 μℓ of thawed plasma is added to the cuvette and mixed by inverting gently up and down for 10 seconds; 20 μℓ of d-ROMs coloring solution Chromogen (N,N diethyl paraphenylenediamine) is used and mixed by gently moving it up and down for 10 seconds in a cuvette. Afterwards, the

cuvette is placed in the thermostat space for 5 minutes and measured with a photometer at 505 nm [11,21].

**Antioxidant capacity (Biological Antioxidant Potential: BAP).**  BAP reacts trivalent iron salt (FeCl3) with colorless thiocyanate to produce a red color as trivalent iron (Fe3+) ions. Then, a sample is added, and the red color is reduced to divalent iron Fe2+ ions by the action of antioxidants in the sample, and the red color is decolorized. The color change is measured with a photometer and the reducing power is evaluated as the antioxidant power. The measured values are μmol/L, indicating the main reactions in the BAP measurement: "FeCl3 + AT (colorless) → [FeCl3-AT (colored)]" and "[FeCl3 + AT (colored)] + BP (e-) → FeCl2 + AT (colorless) + BP". The BAP assay kit cuvette (Thiocyanate HCl) is warmed in the thermostat space set at 37˚C for 10 minutes; 50 μℓ of BAP coloring solution (trivalent iron salt) is taken, added to the cuvette, and mixed gently for 10 seconds. The cuvette is placed in the thermostat space for 3 seconds and measured at 505 nm with a photometer; 10μℓ of thawed plasma is placed in the cuvette and mixed gently for 10 seconds. The cuvette is then placed in the thermostat space and incubated for 5 minutes, and again measured with a 505 nm photometer for 3 seconds [11,21].

## Analysis

The IBM SPSS Statistics ver. 27 was used for analysis to obtain means and standard deviations; the oxidative stress index (OSI) was calculated as "d-ROMs/BAP×100." To examine the characteristics by growth stage, independent t-tests were used for comparisons between the two groups, and the corresponding t-tests were used for pre- and post-experimental comparisons. The percentage change was then calculated for the measured items, where significant differences were found. The confidence interval was set at 95% and the statistical significance level was set at $p < 0.05$.

## Results

### Body weight

Body weight at the end of the experiment was significantly increased by 412.7% ($p < 0.001$) for GPs, 46.3% ($p < 0.001$) for EA, 3.1% ($p < 0.008$) for A, and 0.9% ($p < 0.002$). Conversely, the GPs and EA of the PS group increased significantly by 300.8% and 22.6%, respectively, however, all increases were smaller than those of the C group (all, $p < 0.001$). In contrast, A increased by -12.1% ($p < 0.001$), MA by 12.6% ($p < 0.035$), and OA by -14.6% ($p < 0.001$) (Table 1).

### Wet organ weights

The wet liver weight by growth stage was significantly lighter in all PS groups than in the C group: GPs, -28.4% ($p < 0.001$); EA, -20.3% ($p < 0.001$), A, -17.0% ($p < 0.005$), MA, -20.3% ($p < 0.001$); and OA, -24.1% ($p < 0.001$). However, all growth stages had significantly heavier adrenal wet weights in the PS group than in the C group, with GPs of 20.3% ($p < 0.012$), EA of 20.9% ($p < 0.002$), A of 32.0% ($p < 0.005$), MA of 47.2% ($p < 0.001$), and OA of 22.5% ($p < 0.023$) (Table 2).

### Oxidant stress and antioxidant capacity

At the end of the experiment, d-ROMs were significantly higher in the PS group than in the C group at the four growth stages: GPs, 36.0% ($p < 0.001$), EA, 62.5% ($p < 0.001$), A, 34.9% ($p < 0.001$), and MA, 14.6% ($p < 0.023$). The BAP in the four growth stages was also

**Table 1. Change of body weight (g).**

| Growth stage | Control | | | Psychogenic stress | | |
|---|---|---|---|---|---|---|
| | **before** | **after** | *p-value* | **before** | **after** | *p-value* |
| GPs | 52.8 ± 3.19 (49.5–56.2) | 270.7 ± 10.13 (260.0–281.3) | *0.001* | 53.2 ± 3.87 (49.1–57.2) | 213.2 ± 26.28 (185.6–240.7) | *0.001* |
| EA | 253.8 ± 4.62 (249.0–258.7) | 371.3 ± 14.21 (356.4–386.2) | *0.001* | 254.8 ± 8.13 (246.3–263.4) | 312.5 ± 18.19 (293.4–331.6) | *0.001* |
| A | 440.5 ± 14.64 (425.1–455.9) | 454.3 ± 21.50 (431.3–476.9) | *0.008* | 442.5 ± 15.67 (426.1–458.9) | 389.0 ± 15.30 (372.9–405.1) | *0.001* |
| MA | 548.5 ± 16.67 (531.0–566.0) | 553.5 ± 17.71 (534.9–572.1) | *0.041* | 544.2 ± 12.16 (531.4–556.9) | 475.7 ± 18.77 (456.0–495.4) | *0.001* |
| OA | 624.5 ± 31.26 (591.7–657.3) | 610.3 ± 25.32 (583.8–636.9) | *0.107* | 618.5 ± 24.07 (593.2–643.8) | 528.0 ± 26.63 (500.1–555.9) | *0.001* |

mean ± standard deviation, (): 95% confidence interval.

GPs: Growth periods, EA: Early adulthood, A: Adulthood, MA: Middle-aged, OA: Old-aged.

significantly higher in the PS group than in the C group: EA, 10.7% (p < 0.039), A, 16.9% (p < 0.001), and MA, 32.3% (p < 0.001) (Table 3).

**Growth periods.** During the GPs, d-ROMs increased significantly by 53.0% in the C group and 110.6% in the PS group after the end of the experiment compared to before (all, p < 0.001). The BAP increased significantly by 9.4% in the C group and 19.6% in the PS group (all p < 0.05) (Table 3, Fig 1).

**Early adulthood.** In EA, d-ROMs increased significantly by 54.2% only in the PS group at the end of the experiment compared to the pre-experiment level (p < 0.001). The BAP increased significantly by 14.3% (p < 0.01) in the C group and 25.2% (p < 0.001) in the PS group (Table 3, Fig 1).

**Adulthood.** In A, d-ROMs and BAP increased significantly by 29.1% and 22.0%, respectively (all, p < 0.001) in the PS group (Table 3, Fig 1).

**Middle-aged.** In the MA group, d-ROMs and BAP significantly increased by 15.3% and 20.6%, respectively, in the PS group only after the end of the experiment compared to the pre-experiment level (all, p < 0.05) (Table 3, Fig 1).

**Table 2. The comparison of wet organ weights between the two groups at different growth stages.**

| Growth stage | Liver (g) | | | Adrenal gland (mg) | | |
|---|---|---|---|---|---|---|
| | **Control** | **PS** | *p-value* | **Control** | **PS** | *p-value* |
| GPs | 15.36 ± 0.482 (14.85–15.86) | 11.00 ± 1.855 (9.06–12.95) | *0.001* | 24.63 ± 1.718 (22.83–26.43) | 29.64 ± 3.594 (25.87–33.41) | *0.012* |
| EA | 17.19 ± 0.706 (16.45–17.93) | 13.70 ± 1.099 (12.54–14.85) | *0.001* | 29.60 ± 2.426 (27.04–32.17) | 35.80 ± 2.585 (33.08–38.51) | *0.002* |
| A | 18.28 ± 1.362 (16.85–19.71) | 15.17 ± 1.659 (13.42–16.91) | *0.005* | 23.16 ± 1.985 (21.08–25.24) | 30.57 ± 4.594 (25.75–35.39) | *0.005* |
| MA | 21.76 ± 1.045 (20.66–22.86) | 17.35 ± 0.957 (16.34–18.35) | *0.001* | 24.53 ± 3.074 (21.30–27.75) | 36.10 ± 3.037 (32.91–39.29) | *0.001* |
| OA | 24.84 ± 3.215 (21.46–28.21) | 18.85 ± 0.842 (17.97–19.73) | *0.001* | 38.17 ± 6.863 (30.96–45.37) | 46.75 ± 3.774 (42.79–50.71) | *0.023* |

mean ± standard deviation, PS: Psychogenic stress, (): 95% confidence interval.

GPs: Growth periods, EA: Early adulthood, A: Adulthood, MA: Middle-aged, OA: Old-aged.

Table 3. The comparison between two groups in d-ROM and BAP at different growth stages.

| Growth stage | | Before the experiment | | | After the experiment | | |
|---|---|---|---|---|---|---|---|
| | | Control | PS | p-value | Control | PS | p-value |
| d-ROMs (CARR U) | GPs | 189.2 ± 13.95 (174.5–203.8) | 187.0 ± 19.24 (166.8–207.2) | 0.828 | 289.5 ± 22.55 (265.8–313.2) | 393.8 ± 35.18 (356.9–430.7) | 0.001 |
| | EA | 276.3 ± 21.42 (253.9–298.8) | 282.5 ± 22.33 (259.1–305.9) | 0.636 | 268.2 ± 40.27 (225.9–310.4) | 435.7 ± 17.63 (417.2–454.2) | 0.001 |
| | A | 363.2 ± 38.72 (322.5–403.8) | 366.2 ± 21.36 (343.8–388.6) | 0.871 | 350.5 ± 35.54 (313.2–387.8) | 472.8 ± 37.49 (433.5–512.2) | 0.001 |
| | MA | 355.0 ± 22.92 (330.9–379.1) | 349.8 ± 31.68 (316.6–383.1) | 0.753 | 351.8 ± 22.09 (328.6–375.0) | 403.3 ± 41.29 (360.0–446.7) | 0.023 |
| | OA | 401.8 ± 92.07 (305.2–498.5) | 405.7 ± 57.02 (345.8–465.5) | 0.933 | 365.8 ± 76.95 (285.1–446.6) | 444.2 ± 59.42 (381.8–506.5) | 0.077 |
| BAP (μmol/L) | GPs | 3380.7 ± 137.2 (3236.7–3524.6) | 3266.5 ± 265.3 (2988.1–3544.9) | 0.371 | 3698.3 ± 169.3 (3520.7–3876.0) | 3905.2 ± 371.7 (3515.1–4295.2) | 0.243 |
| | EA | 2854.8 ± 121.1 (2727.7–2981.9) | 2885.8 ± 119.8 (2760.1–3011.6) | 0.665 | 3263.0 ± 315.0 (2932.4–3593.6) | 3612.5 ± 174.6 (3429.2–3795.8) | 0.039 |
| | A | 2753.3 ± 90.24 (2658.6–2848.0) | 2779.0 ± 126.2 (2646.6–2911.4) | 0.694 | 2901.3 ± 257.6 (2631.0–3171.6) | 3391.5 ± 85.68 (3301.6–3481.4) | 0.001 |
| | MA | 2588.8 ± 224.4 (2353.4–2824.3) | 2560.0 ± 139.6 (2413.5–2706.5) | 0.795 | 2735.0 ± 66.73 (2455.1–3014.9) | 3087.8 ± 399.6 (2668.5–3507.2) | 0.102 |
| | OA | 2491.0 ± 447.7 (2021.2–2960.8) | 2449.5 ± 189.9 (2250.2–2648.8) | 0.839 | 2268.3 ± 371.4 (1878.6–2658.0) | 3000.0 ± 305.1 (2979.9–3620.1) | 0.001 |

mean ± standard deviation, PS: Psychogenic stress, (): 95% confidence interval.

GPs: Growth periods, EA: Early adulthood, A: Adulthood, MA: Middle-aged, OA: Old-aged.

**Old-aged.** In the OA group, d-ROMs significantly decreased by -9.0% (p < 0.05) in the C group and significantly increased by 9.5% (p < 0.01) in the PS group at the end of the experiment compared to before the experiment. The BAP increased significantly by 34.7% (p<0.01) in the PS group (Table 3, Fig 1).

**The ratio of oxidative stress to antioxidant capacity (OSI) BAP/d-ROM ratio.** The OSI of the GPs in the C group significantly increased by 40.3% (p < 0.001), and EA significantly decreased by -8.3% (p < 0.05) after the end of the experiment compared to before the experiment. Conversely, the OSI of the PS group showed a significant increase of 77.4% (p < 0.001) for the GPs and 23.4% (p < 0.001) in EA but a significant decrease of -18.7% (p < 0.01) in OA (Fig 2).

## Discussion

In this study, we examined the effects of PS on organisms during five growth stages, from growth to old age. The results revealed that the effects of PS varied depending on the growth stage.

Stress has various negative effects on organisms, causing a decrease in activity and appetite and affecting body weight [22,23]. Body weight markedly increases until A; however, the changes become more gradual after A [24]. Our results on body weight changes in the C group support previous studies [22–24] and revealed significant weight suppression in the PS group. Furthermore, all PS groups showed liver atrophy and adrenal enlargement compared with the C group, indicating PS-induced organ effects. In a study by Sekine et al. [25], PS reduced body weight by 10–20 g, whereas in this study, the range of weight loss in A and beyond was greater. This might be because of the result of differences in the duration of PS used in previous studies [22,25] and in the growth stages of the experimental animals. The PS affects organs related to

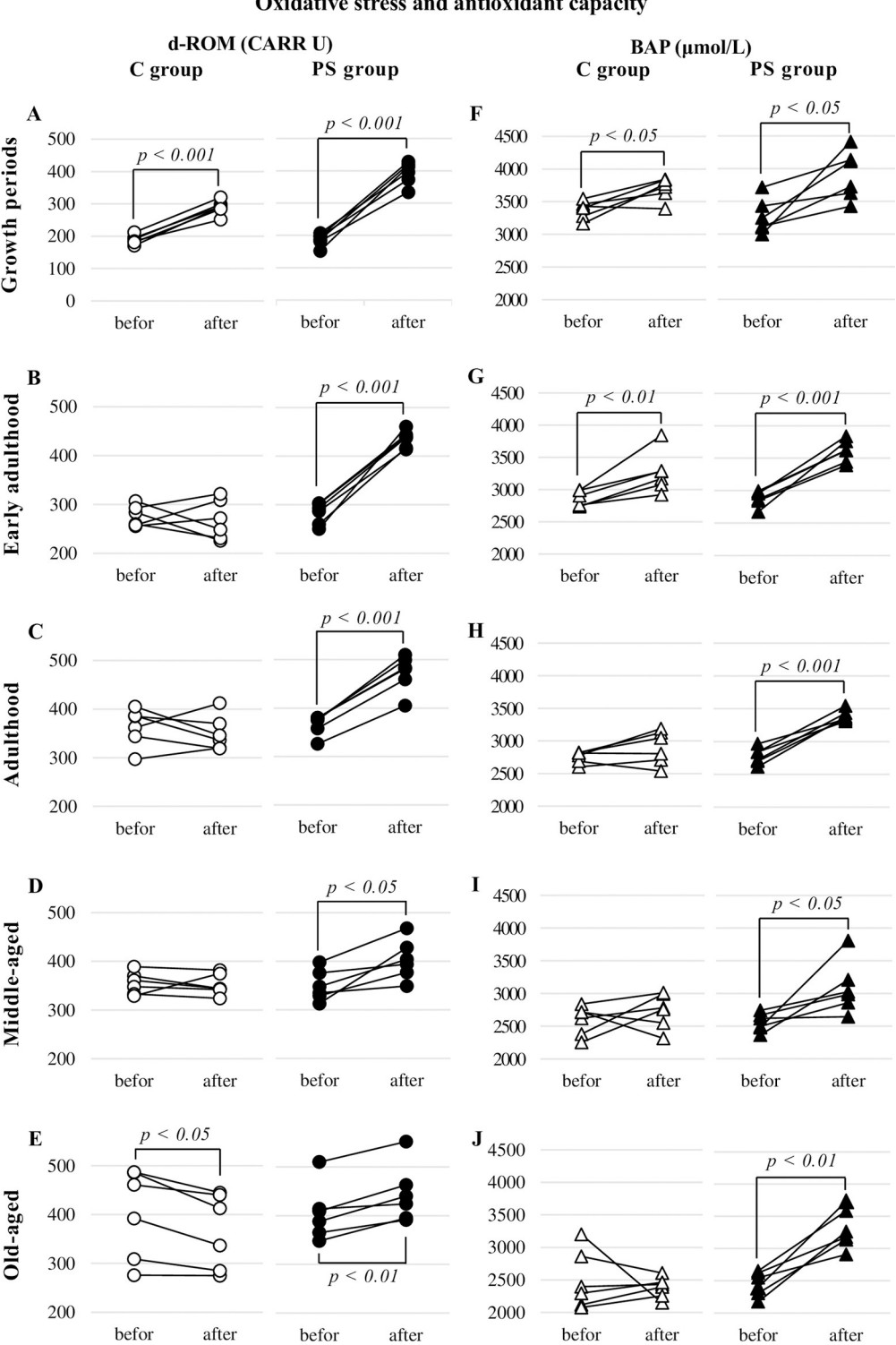

**Fig 1. d-ROM and BAP responses to psychogenic stress.** This figure shows the response of oxidative stress and antioxidant capacity by growth stage to four weeks of psychogenic stress (PS) exposure. Oxidative stress was significantly increased in growth periods (A: p < 0.001) without PS; however, it was significantly decreased in the old-aged rats (E: p < 0.05). In contrast, PS exposure significantly increased in all growth stages (A–C: p < 0.001, D: p < 0.05, E: p < 0.01). Antioxidant capacity was significantly increased in growth periods (F: p < 0.05) and early adulthood (G: p < 0.01) that were not exposed to PS. Contrastingly, PS exposure significantly increased antioxidant capacity in all growth stages (F: p < 0.05; G, H: p < 0.001; I: p < 0.05; J: p < 0.01).

## Oxidative Stress Index

**Fig 2. Response of OSI to psychogenic stress.** The oxidative stress index (OSI) was calculated using the results of oxidative stress and antioxidant capacity (d-ROMs/BAP x 100). The OSI was significantly increased in growth periods ($p < 0.001$) without psychogenic stress; however, it was significantly decreased in EA ($p < 0.05$). Psychogenic stress exposure significantly increased in growth periods and early adulthood (all $p < 0.001$) but decreased significantly in the old-aged rats ($p < 0.01$).

oxidative stress, and there is concern regarding the effects of PS on development during the growth stages [26,27]. In this study, the effects of PS on organs were also shown, with developmental suppression in growing and early adults and atrophy of body composition in A and beyond. In other words, because PS has a greater effect on immature organisms, eliminating PS is a prerequisite for normal development.

Approximately 2% of the oxygen is taken up by an organism is ROS, which fluctuates markedly with hormones and physical activity [11,28]. In addition, about 90% of ROS originate from mitochondria, and increased oxygen uptake [29] and decreased antioxidant function increase oxidative stress in the body [30]. Antioxidant systems that respond to oxidative stress include antioxidant enzymes, such as SOD, GPx, and Catalase (CAT), and non-enzymatic antioxidants, such as ascorbic acid, which are affected by aging [31]. Stress increases corticoid production via the hypothalamic-pituitary-adrenocortical axis and enhances oxidative stress [32–35]. Furthermore, stress affects metabolic pathways related to ROS production, and PS increases oxidative stress even in the absence of physical stimulation [36–38]. Among the five growth stages in this study, the lowest d-ROMs and highest BAPs were observed during the growth phase. In addition, d-ROMs increased in both the C and PS groups after the end of the experiment, however, PS contributed to the further enhancement of d-ROMs and BAP activity. Body composition, including body weight and muscle mass, is highly related to oxygen intake and energy metabolism [39]. The d-ROMs during growth in this study were considered to be the result of changes in oxygen uptake and energy metabolism caused by the development of a pronounced body composition, along with the influence of PS. However, although both the C and PS groups showed a significant increase in body weight, the rate of increase was lower in the PS group than in the C group. These results suggest that PS groups in the growing period may contribute to the suppression of development, increase oxidative stress, and may exert a great burden on immature organisms.

In EA, d-ROMs were significantly elevated only in the PS group, and BAP was significantly elevated in both the C and PS groups. Conversely, the rate of increase in d-ROMs in the PS

group was lower than that in the growing group, however, the rate of increase in BAP was higher than that in the growing group. Du Ruisseau et al. [40] reported a hormonal relationship between the effects of PS on organs and hormones, showing thymic atrophy and adrenal hypertrophy in rats subjected to restraint stress. Kapitonova et al. [41] also showed PS-induced organ atrophy, and Sakamoto et al. [42] found an association between PS-induced immunodeficiency and organs. In other words, the BAP of the C group at a young age is considered a response to antioxidant capacity under the influence of active hormones. In contrast, BAP in the PS group may be the result of PS-induced suppression of organ growth and the effect of PS on hormone-secreting organs. The activity of SOD changes in the order "fetal < newborn < adult" because human dermal fibroblasts have a stronger defense against ROS, which changes in response to the environment [43]. Mousavi et al. [44] reported increased oxidative stress and heart rate in adolescent rats exposed to PS, suggesting that stress during growth could also affect adult rats. In other words, suppressing PS during growth and youth, when biological functions such as immune function are immature, is important as a preventive medicine for good development and aging health.

Although d-ROMs and BAP in A and MA showed a significant increase owing to PS, OSI showed similar results in the C and PS groups, regardless of PS. The ROS greatly influences body composition, including physical activity and muscle mass [10,11,32–35]. However, the rearing environment and changes in body weight in this study suggest that the change in ROS was unlikely to be owing to an increase in activity or muscle mass, but rather a response to PS. Contrarily, the rate of increase in d-ROMs was higher in adult than in middle-aged animals, however, the rate of increase in BAP was similar in both, in the 20% range. Corstjens et al. [45] reported a marked decrease in CAT with age, while Tian et al. [46] reported a decrease in serum levels of SOD, GPx, and CAT. In other words, the pre- and post-experimental OSI in this study was considered to be a specificity of the BAP, reflecting the characteristics of the adult and middle-aged stages with respect to PS.

At rest, before the experiment in the OA group, d-ROMs showed the highest values, and BAP had the lowest values, which were opposite to those in the growth stage. Furthermore, d-ROMs at the end of the experiment differed from those at other growth stages, with a decrease in the C group and a low rate of increase in the PS group. These results suggest that age-related oxidative stress reaches its highest level in old age and at the threshold of resting oxidative stress. In contrast, in the PS group, BAP showed an excess response of 34.7% to 9.5% increase in d-ROMs. Rizvi et al. [31] suggested that the induction of antioxidant enzyme activity during aging is a compensatory response to the dynamics of oxidative stress. In the aging stage of this study, the standard deviation values of d-ROMs and BAP at rest in the C group were significantly larger than those in the other growth stages. Based on these results, we supported the characteristics of antioxidant capacity shown by Rizvi et al. [31] and considered the results to be the influence of individual characteristics on old age. In other words, the individual differences in the body in old age may have contributed to the changes in OSI by causing fluctuations in the hyperactive state of the ROS-generating and antioxidant systems.

The higher the response to oxidative stress, the longer the life span in nature [47], and antioxidant capacity plays an important role in pathological conditions [48]. Therefore, when examining the oxidative stress and antioxidant capacity that affect disease and health, it is necessary to consider the characteristics of the GP, when the body is rapidly developing, and the older age period, when individual differences are high.

The study reached its basic findings by investigating both the biochemical effects of psychogenic stress on oxidative stress and antioxidant capacity at different ages and the morphological effects on body weight and organs. The PS and age characteristics of the present study's subjects can be considered to be fundamental data that may make the study adaptable to

humans. However, one limitation of this study is that we were unable to examine the effects of gender characteristics and diet on oxidative stress and antioxidant capacity. These will be the subject of the next study.

## Conclusion

The biological response to PS reflects the body's developmental status and age characteristics, with a particularly high negative impact during the GP. In addition, in older age, when individual differences in the body are high, the response of excessive antioxidant capacity to increased oxidative stress owing to PS is exhibited. Therefore, suppression of PS with respect to age contributes to good body growth and stable oxidative stress levels.

## Acknowledgments

We would like to express my sincere gratitude to the Professor Takao Suzuki whose comments and suggestions were of tremendous value for my study.

## Author Contributions

**Conceptualization:** Sangun Lee, Haruka Tatehana.

**Data curation:** Atsunori Itagaki, Atsuko Satoh.

**Formal analysis:** Atsunori Itagaki, Issei Sugimoto, Haruka Tatehana.

**Funding acquisition:** Sangun Lee.

**Investigation:** Issei Sugimoto, Takumi Saito.

**Methodology:** Yoshihiko Shibukawa.

**Software:** Takumi Saito.

**Validation:** Yoshihiko Shibukawa.

**Writing – original draft:** Sangun Lee.

**Writing – review & editing:** Sangun Lee, Yoshihiko Shibukawa.

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
