## [Decision Letter · Decision Letter 0]

24 Aug 2023

PONE-D-23-17434Effects of psychogenic stress on oxidative stress and antioxidant capacity at different growth stages.PLOS ONE

Dear Dr. LEE,

Thank you for submitting your manuscript to PLOS ONE. After careful consideration, we feel that it has merit but does not fully meet PLOS ONE’s publication criteria as it currently stands. Therefore, we invite you to submit a revised version of the manuscript that addresses the points raised during the review process.

We look forward to receiving your revised manuscript.

Kind regards,

Yasmina Abd‐Elhakim

Academic Editor

PLOS ONE

Journal Requirements:

4. Please include your tables as part of your main manuscript and remove the individual files. Please note that supplementary tables (should remain/ be uploaded) as separate "supporting information" files

Reviewers' comments:

Reviewer's Responses to Questions

**Comments to the Author**

1. Is the manuscript technically sound, and do the data support the conclusions?

Reviewer #1: Yes

Reviewer #2: Yes

Reviewer #3: No

Reviewer #4: Yes

2. Has the statistical analysis been performed appropriately and rigorously? 

Reviewer #1: No

Reviewer #2: Yes

Reviewer #3: Yes

Reviewer #4: I Don't Know

3. Have the authors made all data underlying the findings in their manuscript fully available?

Reviewer #1: Yes

Reviewer #2: No

Reviewer #3: No

Reviewer #4: Yes

4. Is the manuscript presented in an intelligible fashion and written in standard English?

Reviewer #1: Yes

Reviewer #2: Yes

Reviewer #3: Yes

Reviewer #4: Yes

5. Review Comments to the Author

Reviewer #1: I assume the ethical approval has been obtained prior to this study.

A very informative piece of study, explaining the effects of the relevant factors in the process , therefore making this research applicable to various and different scientific subjects such as biology, medicine, psychology, neurodevelopmental science, aging and even the field of cosmetic science.

I found it relevant and useful in the field of neurodevelopmental sciences and effects on different stages of life.

Looking forward to more helpful studies by these authors as long as the ethical approval is secured.

Reviewer #2: Effects of psychogenic stress on oxidative stress and antioxidant capacity at different growth stage

This is an important study that addresses the effect of psychogenic stress on the growth of an organism. Results of this study show that effects vary with the growth stage

1. The results presented are original

2. Results presented in pre-print

.

3. Experiments are satisfactory done

4. Conclusions are in keeping with the data

5. The sufficiently presented in Standard English and intelligible fashion

6. Ethical approval was obtained

.

General comments

The title seems incomplete. I suggest the authors to rewrite as

experimental study of the Effects of psychogenic stress on oxidative stress and antioxidant capacity at different growth stages of rats

Under materials and methods, the subtitle “materials is” may be not appropriate for the experimental animals. I suggest that it is removed. The weight of the animals should be included in this section. Therefore the subtitle of body weight may be removed and remain with the subtitle blood collection instead of” blood draw”

Under blood collect include the amount of blood that was collected

In the methods section under the sub title of ““Organ evisceration” the sentence the liver and adrenal were removed seems to be incomplete. Was it placed in a any solution before analysis?

Under methods in the analysis section include the set confidence interval

What is the strength of this study?

Include limitations to this study

What are the recommendations of the study?

Reviewer #3: The study by Dr Lee et al describes of changes in reactive oxygen metabolites/species and anti-oxidant potential/capacity in the serum of rats exposed to strong psychogenic stress at different life stages.

The study addresses an important question a show the bodily stressors connect to cellular stress responses.

However, the methods are not precisely described and the measurement of somehow too general.

For instance, the RWIS procedure is not described in detail. As the size of the animals is increasing with the age it is also necessary to give the dimensions of the restrainers used throughout the experiments. It is also not mentioned how the control animals were maintained and were they restrained for the blood withdrawal. The specific details of the anesthetic mixture and the precise dose should also be given. The details of blood treatment should also be described.

There are much more updated, state-of-the-art methods to specify reactive oxygen species which are respond to psychogenic stress.

Finally, changes in body and adrenal indicate the success of the stress exposure, direct measurement of stress hormones (corticosterone, ACTH) and the relation between d-ROM BAP and inflammatory mediators would have been highlighted the significance of the study.

Reviewer #4: The study is an original experimental research. The article was written in standard English, and precise language was used. Experimental results are explained in the manuscript by comparing them with the results of the studies in the literature. Experimental groups (control and others) are sufficient in number. The experimental protocol (RWIS method), as well as the adequacy of the sample size, can be explained in more detail.

The source protocols stated that the subjects were immersed in water after 24 hours of fasting. In this study, were the experimental animals (PS group) fast before being immersed in water? Is 24-hour fasting included in the PS protocol?

If immersed in water after 24 hours of fasting, it would be appropriate to explain the effect of this fasting period on subjects in the growth phase.

6. PLOS authors have the option to publish the peer review history of their article (what does this mean?). If published, this will include your full peer review and any attached files.

Reviewer #1: **Yes: **Dr Lily Abedipour MD

Reviewer #2: **Yes: **ESTER LILIAN ACEN

Reviewer #3: No

Reviewer #4: **Yes: **Serpil ERŞAN

---

## [Author Response · Author response to Decision Letter 0]

25 Oct 2023

Reviewer #1: I assume the ethical approval has been obtained prior to this study.

A very informative piece of study, explaining the effects of the relevant factors in the process , therefore making this research applicable to various and different scientific subjects such as biology, medicine, psychology, neurodevelopmental science, aging and even the field of cosmetic science.

I found it relevant and useful in the field of neurodevelopmental sciences and effects on different stages of life.

Looking forward to more helpful studies by these authors as long as the ethical approval is secured.

Reviewer #2: Effects of psychogenic stress on oxidative stress and antioxidant capacity at different growth stage

This is an important study that addresses the effect of psychogenic stress on the growth of an organism. Results of this study show that effects vary with the growth stage

1. The results presented are original

2. Results presented in pre-print

.

3. Experiments are satisfactory done

4. Conclusions are in keeping with the data

5. The sufficiently presented in Standard English and intelligible fashion

6. Ethical approval was obtained

.

General comments

The title seems incomplete. I suggest the authors to rewrite as

experimental study of the Effects of psychogenic stress on oxidative stress and antioxidant capacity at different growth stages of rats

Under materials and methods, the subtitle “materials is” may be not appropriate for the experimental animals. I suggest that it is removed. The weight of the animals should be included in this section. Therefore the subtitle of body weight may be removed and remain with the subtitle blood collection instead of” blood draw”

Under blood collect include the amount of blood that was collected

In the methods section under the sub title of ““Organ evisceration” the sentence the liver and adrenal were removed seems to be incomplete. Was it placed in a any solution before analysis?

Under methods in the analysis section include the set confidence interval

What is the strength of this study?

Include limitations to this study

What are the recommendations of the study?

Reviewer #3: The study by Dr Lee et al describes of changes in reactive oxygen metabolites/species and anti-oxidant potential/capacity in the serum of rats exposed to strong psychogenic stress at different life stages.

The study addresses an important question a show the bodily stressors connect to cellular stress responses.

However, the methods are not precisely described and the measurement of somehow too general.

For instance, the RWIS procedure is not described in detail. As the size of the animals is increasing with the age it is also necessary to give the dimensions of the restrainers used throughout the experiments. It is also not mentioned how the control animals were maintained and were they restrained for the blood withdrawal. The specific details of the anesthetic mixture and the precise dose should also be given. The details of blood treatment should also be described.

There are much more updated, state-of-the-art methods to specify reactive oxygen species which are respond to psychogenic stress.

Finally, changes in body and adrenal indicate the success of the stress exposure, direct measurement of stress hormones (corticosterone, ACTH) and the relation between d-ROM BAP and inflammatory mediators would have been highlighted the significance of the study.

Reviewer #4: The study is an original experimental research. The article was written in standard English, and precise language was used. Experimental results are explained in the manuscript by comparing them with the results of the studies in the literature. Experimental groups (control and others) are sufficient in number. The experimental protocol (RWIS method), as well as the adequacy of the sample size, can be explained in more detail.

The source protocols stated that the subjects were immersed in water after 24 hours of fasting. In this study, were the experimental animals (PS group) fast before being immersed in water? Is 24-hour fasting included in the PS protocol?

If immersed in water after 24 hours of fasting, it would be appropriate to explain the effect of this fasting period on subjects in the growth phase.

---

## [Decision Letter · Decision Letter 1]

4 Jan 2024

PONE-D-23-17434R1Effects of psychogenic stress on oxidative stress and antioxidant capacity at different growth stages of rats: Experimental studyPLOS ONE

Dear Dr. LEE,

Thank you for submitting your manuscript to PLOS ONE. After careful consideration, we feel that it has merit but does not fully meet PLOS ONE’s publication criteria as it currently stands. Therefore, we invite you to submit a revised version of the manuscript that addresses the points raised during the review process.

We look forward to receiving your revised manuscript.

Kind regards,

Yasmina Abd‐Elhakim

Academic Editor

PLOS ONE

Journal Requirements:

**Additional Editor Comments:**

The authors need to address the reviewer's comments and the following points carefully:

A more detailed description of the methods, like d-ROM and BAP measurement, should be added.

English editing of the manuscript is highly needed as several typographic, grammatical, and overall style errors still exist.

The composition of the anesthetic should be described as the concentration of the anesthetic agent and then as mg/kg. 

In the conclusion section: ""investigating both the morphological and biochemical effects". Clarify what you meant by morphological effects.

Reviewers' comments:

Reviewer's Responses to Questions

**Comments to the Author**

1. If the authors have adequately addressed your comments raised in a previous round of review and you feel that this manuscript is now acceptable for publication, you may indicate that here to bypass the “Comments to the Author” section, enter your conflict of interest statement in the “Confidential to Editor” section, and submit your "Accept" recommendation.

Reviewer #2: All comments have been addressed

Reviewer #3: (No Response)

Reviewer #5: All comments have been addressed

2. Is the manuscript technically sound, and do the data support the conclusions?

Reviewer #2: Yes

Reviewer #3: Partly

Reviewer #5: (No Response)

3. Has the statistical analysis been performed appropriately and rigorously? 

Reviewer #2: Yes

Reviewer #3: No

Reviewer #5: (No Response)

4. Have the authors made all data underlying the findings in their manuscript fully available?

Reviewer #2: Yes

Reviewer #3: (No Response)

Reviewer #5: (No Response)

5. Is the manuscript presented in an intelligible fashion and written in standard English?

Reviewer #2: Yes

Reviewer #3: No

Reviewer #5: (No Response)

6. Review Comments to the Author

Reviewer #2: The manuscript has been greatly improved on. All comments have been addressed clearly and I recommend publication

Reviewer #3: The manuscript still needs revision.

Please correct the typos.

Give a more precise description on the measurement of d-ROMs and BAP.

Describe the "morphological" effects of stress...(as it is stated in the discussion)

Anesthetic composition is not described correctly. Use concentration instead of volumes.

Consider a discussion on age-dependent changes in all measures.

Reviewer #5: (No Response)

7. PLOS authors have the option to publish the peer review history of their article (what does this mean?). If published, this will include your full peer review and any attached files.

Reviewer #2: **Yes: **Ester Lilian Acen

Reviewer #3: No

Reviewer #5: **Yes: **Amany Abdel-Rahman Mohamed

---

## [Author Response · Author response to Decision Letter 1]

14 Feb 2024

PONE-D-23-17434R1

Effects of psychogenic stress on oxidative stress and antioxidant capacity at different growth stages of rats: Experimental study

Additional Editor Comments:

A more detailed description of the methods, like d-ROM and BAP measurement, should be added.

➤Correction

The assay method was as follows: The d-ROMs assay kit cuvette (4.8PH acetate buffer) is placed in a thermostat space set at 37°C and allowed to warm for 10 minutes. Then, 20 µℓ of thawed plasma is added to the cuvette and mixed by inverting gently up and down for 10 seconds; 20 µℓ of d-ROMs coloring solution Chromogen (N,N diethyl paraphenylenediamine) is used and mixed by gently moving it up and down for 10 seconds in a cuvette. Afterwards, the cuvette is placed in the thermostat space for 5 minutes and measured with a photometer at 505 nm [11,21].

➤Correction

The BAP assay kit cuvette (Thiocyanate HCl) is warmed in the thermostat space set at 37°C for 10 minutes; 50 µℓ of BAP coloring solution (trivalent iron salt) is taken, added to the cuvette, and mixed gently for 10 seconds. The cuvette is placed in the thermostat space for 3 seconds and measured at 505 nm with a photometer; 10µℓ of thawed plasma is placed in the cuvette and mixed gently for 10 seconds. The cuvette is then placed in the thermostat space and incubated for 5 minutes, and again measured with a 505 nm photometer for 3 seconds [11,21].

English editing of the manuscript is highly needed as several typographic, grammatical, and overall style errors still exist.

➤Correction

The composition of the anesthetic should be described as the concentration of the anesthetic agent and then as mg/kg. 

➤Correction

The anesthesia was prepared with 25 ml of three types of mixed anesthetic agents (Medetomidine Hydrochloride 7.5µg; 0.375 mg/kg + Midazolam 0.4 mg; 2.0 mg/kg + Butorphanol Tartrate 0.5 mg;2.5 mg/kg) and administered intraperitoneally (0.5 ml / BW100 g) after blood collection at the end of the experiment.

In the conclusion section: ""investigating both the morphological and biochemical effects". Clarify what you meant by morphological effects.

➤Correction

The study reached its basic findings by investigating both the biochemical effects of psychogenic stress on oxidative stress and antioxidant capacity at different ages and the morphological effects on body weight and organs.

---

## [Editor Report · Decision Letter 2]

21 Feb 2024

Effects of psychogenic stress on oxidative stress and antioxidant capacity at different growth stages of rats: Experimental study

PONE-D-23-17434R2

Dear Dr. LEE,

We’re pleased to inform you that your manuscript has been judged scientifically suitable for publication and will be formally accepted for publication once it meets all outstanding technical requirements.

Kind regards,

Yasmina Abd‐Elhakim

Academic Editor

PLOS ONE
---

## [Editor Report · Acceptance letter]

2 Apr 2024

PONE-D-23-17434R2 

PLOS ONE

Dear Dr. Lee , 

I'm pleased to inform you that your manuscript has been deemed suitable for publication in PLOS ONE. Congratulations! Your manuscript is now being handed over to our production team.

Kind regards, 

on behalf of

Prof. Dr. Yasmina Abd‐Elhakim 

Academic Editor

PLOS ONE